# Statistical Methods for Quality Equivalence of Topical Products. 0.5 mg/g Betamethasone Ointment as a Case-Study

**DOI:** 10.3390/pharmaceutics12040318

**Published:** 2020-04-02

**Authors:** Jordi Ocaña, Toni Monleón-Getino, Virginia Merino, Daniel Peris, Lluís Soler

**Affiliations:** 1Department of Genetics, Microbiology and Statistics, Section of Statistics, University of Barcelona, Av. Diagonal, 643, 08028 Barcelona, Spain; jocana@ub.edu; 2Departamento de Farmacia y Tecnología Farmacéutica y Parasitología, Facultad de Farmacia, Universitat de València, Av. Vicente Andrés Estellés s/n, Burjassot, 46100 Valencia, Spain; virginia.merino@uv.es; 3Instituto Interuniversitario de Investigación de Reconocimiento Molecular y Desarrollo Tecnológico (IDM), Universitat Politècnica de València, Universitat de València, 46100 Valencia, Spain; 4Department of Clinical Affairs & Development. Uriach, Pol. Ind. Riera de Caldes, Avda. Camí Reïal 51-57, 08184 Palau-Solitá i Plegamans, Spain; daniel.peris@uriach.com; 5Department of Formulation and Late Scale Development, Kern Pharma. Av Venus 72, Pol. Ind Colon II, 08228 Terrassa, Spain; lsoler@kernpharma.com; 6GRBIO, Research Group in Biostatistics and Bioinformatics, 08028 Barcelona, Spain; 7BIOST3, Research Group in Clinical Statistics, Bioinformatics and Computational Biodiversity, 08028 Barcelona, Spain

**Keywords:** Fieller’s confidence interval, equivalence test, batch effect, multiple quality variables, principal component analysis

## Abstract

This study examines the statistical implications, and their possible implementation, of the “Draft guideline on quality and equivalence of topical products” issued by the European Medicines Agency in 2018, with particular focus on the section devoted to quality equivalence of physical properties. A new confidence interval to conduct the quality equivalence test and a way to cope with the multiplicity of quality parameters are presented and discussed. As an example, the results and the statistical analysis of a study on betamethasone 0.5 mg/g ointment are presented. It is suggested that the equivalence limits proposed in the draft guideline are overly strict: It is as difficult to declare quality equivalence between two packaging formats of the same reference product as to declare quality equivalence between the reference and the test product.

## 1. Introduction

In October, 2018, the European Medicines Agency (EMA) issued a draft guideline on the quality and equivalence of topical products [1]. On line 630 in Section 5.2.1, “Extended pharmaceutical equivalence acceptance criteria”, it is stated that: “For quantitative quality characteristics, the 90% confidence interval for the difference of means of the test and comparator products should be contained within the acceptance criteria of +/−10% of the comparator product mean, assuming normal distribution of data”. The current study focusses on the consequences of the draft guideline in general, and this statement in particular, from a statistical point of view. The main objectives of the study were: (i) To ascertain which experimental design is the most appropriate; (ii) to construct adequate testing procedures to prove quality equivalence between test and reference products; (iii) to elucidate how to manage the great multiplicity of available quality characteristics and (iv) to shed light on the chances of finally declaring equivalence under this regulatory proposal, even in an scenario of reasonably real equivalence. In other words, we asked the question: are these criteria too restrictive? In order to illustrate these ideas, we applied a dataset using 0.5 mg/g betamethasone ointment as a case-study. Some additional comments about these objectives are set forth below.

The guideline [1] states that variability between batches of product must be considered. This is a very reasonable demand because the data gathering process involves separately obtaining batches of the reference and test products and then determining the required quality measures separately for each of these batches. All subsequent theoretical developments and statistical analyses must be done under the framework of the most appropriate experimental design modelling this process of data collection.

The statement quoted at the start of the Introduction enunciates the most frequent approach to equivalence testing (here in its two-sided version), the principle of confidence interval inclusion: Equivalence can be declared if the confidence interval for the parameter of interest lies within the equivalence limits. The problem is thereby reduced to finding an appropriate confidence interval for the parameter of interest. Here, according to the draft guideline (“… the difference of means of the test and comparator products … of the comparator product mean …”), the parameter of interest is the difference between the population means of the test and the reference products, say *µ_T_* and *µ_R_* respectively, relative to the reference product, (*µ_T_* − *µ_R_*)/*μ_R_*. As is proven here (see Methods), the equivalence problem for (*µ_T_* − *µ_R_*)/*μ_R_* can be reduced to an equivalence problem for the ratio *µ_T_*/*µ_R_*. In this study a new, exact, confidence interval for this ratio is introduced. It should also be noted that a direct difference (*µ_T_* − *µ_R_*) would be expressed in the unit of measurement in which each quality parameter itself is expressed, with very different scales of measurement. On the other hand, the percentages defining the acceptance criterion are unit-less quantities, in accordance with these mean ratios.

As has been stated, once the confidence interval has been obtained, it must be fully included in the equivalence limits in order to declare equivalence. The preliminary regulatory proposal of equivalence limits is ±0.1, or ±10 in terms of percentages, but it may be worth considering less restrictive limits, e.g., similar to those in bioequivalence/bioavailability studies.

In “classical” bioequivalence studies, there is common agreement on what must be proved similar between the test and the reference formulation: The bioavailability of the active principle at its site of action. There is also broad consensus on measuring bioavailability by means of two main variables for single dose studies: *Cmax*, the maximum active principle concentration reached, and *AUC*, the area under the curve, both obtained from the curve of active principle concentration in plasma vs. time after the administration. On the other hand, in topical equivalence, there are many ways to measure the performance of a semisolid product, ranging from the qualitative and quantitative composition to microstructure/physical properties or in vitro release. All together are considered by the guideline as equivalence with respect to quality or extended pharmaceutical equivalence. The physical properties include density, rheological properties (e.g., thixotropy and viscosity, among many others), and pH. As it is nearly impossible to declare equivalence for all these potential measures of quality, it would seem reasonable to restrict the number of conceptual categories required to demonstrate equivalence, and in the multivariate (e.g., rheology), to summarize the information in only one or a few measures. An obvious way would be to give weights to the quality measures (after coming to a previous agreement on which of them should be measured) and to summarize them by means of a linear combination based on these weights. One possibility is to draw up a regulation that would fix in advance these measures and their weights. As this is not within our reach of expertise, here we discuss an alternative possibility based on the standard multivariate analysis technique of principal components analysis, PCA.

Admittedly, an underlying objective of this work is to demonstrate that the current regulatory proposal is too stringent. In this regard, using the betamethasone data, we have tried to answer the following question: Which is harder, to demonstrate equivalence between two variants of the reference product or to demonstrate equivalence between test and reference products?

## 2. Materials and Methods

In this paper we present a quality equivalence study of physical properties, comparing a new topical product developed by Kern Pharma (test product) equivalent to the reference Leo Pharma Daivodet reference product. For further comparison, this study is complemented with a quality equivalence analysis comparing the reference batches themselves.

The physico-chemical properties of batches of an ointment formulation containing 0.5 mg/g betamethasone prepared by Kern Pharma were compared with those of reference formulation batches (Daivobet). The 12 batches under study were the following: Test (Betamethasone Kern): G1802264, G180001 and Reference (Daivodet): A66014, A51886, A57973, A51002, A58066, A65328, A53211, A45843, A56687, A67822 [Daivodet 60g: A66014, A51886, A57973, A51002, A58066; Daivodet 30g: A65328, A53211, A45843, A56687, A67822]. The evaluated properties include rheology and spreadability. All physico-chemical analyses were performed at the Department of Pharmacy and Pharmaceutical Technology and Parasitology of the University of Valencia.

The quality variables considered in this study are described below: 

### 2.1. Rheological Analysis

The tests were performed with a controlled stress rheometer (RheoStress RS1, Thermo Haake^®^, Karlsruhe, Germany) connected to a Haake K10 thermostatic bath, using the RheoWin 4.0 software (Karlsruhe, Germany). A serrated parallel plate (35 mm diameter, 1 mm gap) was used. Measurements were done at 25 °C, after allowing samples to relax for 600 s. All rheological parameters were calculated using the Kaleidagraph 4.03 software (Synergy Software, Reading (PA), USA). The illustrative computations are based on the following rheological parameters:Relative thixotropy (*Sr*);yield stress (*σ*_0_);zero shear viscosity (*η*_0_);viscosity at 100 s^−1^ (*η*_100_);viscoelastic storage or modulus (*G′*);loss or viscous modulus (*G″*); andloss tangent (tan *δ*).

#### 2.1.1. Hysteresis Loops

Setting the rheometer in controlled shear rate mode, hysteresis loops were performed in three stages: (1) Upward curve, measuring from 1 s^−1^ and 100 s^−1^ in a stepped ramp; (2) continuous rotation at 100 s^−1^ for 60 s; and (3) downward curve, measuring from 100 s^−1^ to 1 s^−1^ in a stepped ramp.

Both the area enclosed by the upward (*S_A_*) and the downward (*S_D_*) curves were calculated by numerical integration, the difference between them being the thixotropic area (*S_T_*). Finally, the relative thixotropic areas were obtained using Equation (1), [2], and were used for comparison:(1)$$S_{R}(\%)=\frac{S_{A}-S_{D}}{S_{A}}\times 100$$

#### 2.1.2. Flow Curves

Setting the rheometer in controlled stress mode, stepped flow curves between 1 Pa (initial *σ*) and 1300 Pa (final *σ*) were performed. The simplified Carreau model was fitted to viscosity values as a function of shear rate data (*γ*) to calculate *η*_0_, the zero-shear viscosity, using Equation (2), [3]:(2)$$\eta =\frac{\eta_{0}}{{\left[1+\;{\left(\frac{\dot{\gamma }}{{\dot{\gamma }}_{c}}\right)}^{2}\right]}^{s}}$$
where, ${\dot{\gamma }}_{C}$ is the critical shear rate, and *s* the shear thinning index. Viscosity corresponding to 100 *s*^−1^ (*η*_100_) was also calculated from the curve fits obtained.

Yield stress *σ*_0_ was estimated on the double log scaled rheograms *σ*_0_
*= f* (*σ*) from the point where the straight line corresponding to the viscosity plateau intersects the tangent line to the fall in viscosity [4].

#### 2.1.3. Viscoelastic Properties from Oscillatory Tests

The elastic (*G′*) and viscous (*G″*) moduli were obtained from the frequency sweep tests performed between 0.01 and 10 Hz at a stress amplitude of 10 Pa (within the linear viscoelastic region, previously determined from stress sweep tests at 1 Hz).

The observed moduli at a frequency of 1 Hz, as well as the tangent of the phase shift, or loss tangent, which gives the relation between both dynamic moduli (3), also calculated at 1 Hz, were used for comparison, as Equation (3) indicates.
(3)$$\tan\delta =\frac{G^{\prime \prime }}{G^{\prime }}$$

### 2.2. Spreadability Measurements

Spreadability was calculated by progressive addition of 50, 100, and 200 g weights on a glass dish placed on 1 g of sample (10 batches; 12 replicates per batch). The surface was measured after 5 minutes. The area under the surface *versus* weight curve was calculated using Excel 2016 (Microsoft).

### 2.3. Usefulness of These Parameters as Representative of the Internal Structure of Semisolids

Semisolid pharmaceutical formulations have a complex structure, related with their rheological properties, that can be characterized using continuous rotational and oscillatory tests. Most of those semisolids are pseudoplastic, that is, their viscosity varies with the shear rate applied. Continuous rotational tests provide information about the flux properties of a pharmaceutical system as yield stress, viscosity and thixotropy. 

Apparent yield stress (*σ*_0_) corresponds to the strength that must be applied so that the system begins to decrease in viscosity. This value represents the force required to extract the formulation from the tube, or to start applying it easily on the skin.

Zero shear viscosity (*η*_0_), the viscosity of the formulation at very small shear rates, is related with its consistency at rest, in the container. On the other hand, since viscosity decreases when applying higher shear rates, it is interesting to know the viscosity at rubbing speeds, that is, those that correspond to the application on the skin. Nevertheless, these speeds are very high (they may be greater than 1000 s^−1^) and can cause spinning of the sample in the rheometer. For this reason, the viscosity at 100 s^−1^ (*η*_100_) was selected as a comparative estimate. 

If the viscosity decreases when a constant shear is applied for a period of time, it is indicative that the fluid is thixotropic. Although the thixotropy phenomenon really implies the recovery of the sample, it is of interest to characterize the reodestruction produced by agitation over time. The behavior of the material was characterized by the relative thixotropic area, *S_R_*, which indicates the extent of the broken structure during the time of the experiment.

All these properties are important in the development and in the quality control of formulations, as the viscosity and yield stress affect organoleptic properties, including product spreadability. Suspensions that present thixotropic behavior have good physical stability and the behavior of semi-solid formulations in mixing and transfer operations between containers can be anticipated if their flow properties are known.

On the other hand, dynamic oscillation testing is a powerful tool that reveals the microscopic structure of a viscoelastic material. Storage and loss moduli (*G*′ and *G*″) and their dependence on frequency oscillation provide information about viscoelastic properties. If *G*′ > *G*″, elastic behavior predominates. If both moduli depend on frequency, and are more or less parallel, in a log–log plot, the systems have a gel structure (strong if *G*′ in independent of frequency and the distance between *G*′ and *G*″ is big, or weak if *G*′ is only slightly dependent on the frequency and the distance between both parameters is small). However, if the loss modulus is greater than the storage module (*G*″ > *G*′) and both are dependent on frequency, behavior is predominantly viscous. These systems are called macromolecular solutions, as the internal structure of the system is due to the crosslinking between molecular chains and not to a three-dimensional network. The ratio of both moduli is given by the loss tangent (tan *δ*), so low values of the tangent correspond to substances in which the elastic behavior predominates over the viscous, with a defined internal structure. 

In all the semisolid products tested, the predominance of elastic behavior was clear (*G*′ > *G*″) and both moduli showed variation with frequency.

The last parameter calculated, the spreadability index (*AUC*), is used as a routine control for semisolid preparations and provides information about the spreadability of a formulation when applied on skin, although in a simpler manner that the more rigorous analysis of viscosity.

### 2.4. Statistical Theory and Methods

#### 2.4.1. Experimental Design

The most appropriate experimental design describing the process of acquiring data is a two-factors design, factor “form”, fixed, with *a* = 2 levels, “Test” and “Reference” (*T* and *R* from now on), and random factor “batch”, hierarchically nested into form, with *b_T_* levels (batches) for form *T* and *b_R_* for form *R*. Considering that the availability of batches may differ according to the form, *b_T_* and *b_R_* may differ also. For each batch, we may reasonably assume a balanced, constant number of replicate observations, *n*. (For simplicity we present here the formulae for balanced *n*; the unbalanced case conducts to more complicated expressions but it is still affordable.) This design choice is further justified as follows. It is clear that in a test vs. reference equivalence study, the main objective is to study the “form” factor and that the fixed levels to consider are precisely those: *T* and *R*. On the other hand, factor “batch” is nested to “form” because the selected test batches are different from the selected reference batches. The levels of “batch” (i.e., the specific batches under study) are just a sample of all the possible batches that can be potentially studied. So it is suitable to consider “batch” as a random factor.

Between-batch and within-batch variabilities should be managed within the framework of this hierarchical design, which involves well-known standard statistical theory (see, for example, Kuehl [5], Section 5.9). The main results are included here as they are indispensable to follow the developments in next subsections.

The linear model associated with this design may be written as in Equation (4):(4)$$Y_{ijk}=\mu +\phi_{i}+B_{j\left(i\right)}+e_{k\left(ij\right)}$$
where *Y_ijk_* stands for the observed response (e.g., quality measures like the viscosity, or the area under the curve measuring spreadability, or a PCA summary of the rheology variables, as is discussed below); *μ* stands for the constant or global model mean; the constant *ϕ**_i_* corresponds to the fixed effect of the i-th formulation, *i* = *T*, *R*; *B_j_*_(*i*)_ are random variables specifying the random effect of the j-th batch of the i-th formulation; and *e_k_*_(*ij*)_ stands for the residual associated with the k-th replicate observation for the j-th batch of the i-th formulation. As is usual, we assume that all the *B_j_*_(*i*)_ and the *e_k_*_(*ij*)_ are mutually independent, with a mean of zero. The common “batch” (between-batch) variance will be designated $\sigma_{B}^{2}$ and the common residual or within-batch variance as $\sigma_{W}^{2}$.

The mean response for each formulation, *µ_T_* = *µ* + *ϕ**_T_* and *µ_R_* = *µ* + *ϕ**_R_*, may be unbiasedly estimated by the formulae in Equations (5) and (6):(5)$${\bar{Y}}_{i}=\frac{1}{b_{i}}\sum_{j=1}^{b_{i}}\left\{\frac{1}{n}\sum_{k=1}^{n}Y_{ijk}\right\}\;\mathrm{for}\;i=T,R.$$
with variance:(6)$$\left(n\sigma_{B}^{2}+\sigma_{W}^{2}\right)/\left(nb_{i}\right)\;\mathrm{for}\;i=T,R.$$

These variances depend on both the within-batch and the between-batch variance and in general are greater than those ignoring the batch effect. In other words, ignoring the batch effect would conduct to an erroneous and unfair testing procedure favoring the equivalence declaration.

The batch mean square error in the ANOVA table, *MS_B_*, has the expectation, Equation (7):(7)$$E\left(MS_{B}\right)=n\sigma_{B}^{2}+\sigma_{W}^{2}.$$

Thus, the variances in (6) may be estimated as in Equation (8):(8)$$\hat{\mathrm{var}}\left({\bar{Y}}_{T}\right)=\frac{MS_{B}}{nb_{T}}\;\mathrm{and}\;\hat{\mathrm{var}}\left({\bar{Y}}_{R}\right)=\frac{MS_{B}}{nb_{R}}.$$

#### 2.4.2. Testing for Quality Equivalence. Ratio of Means Confidence Intervals

##### Direct Confidence Interval for the Ratio

As has been stated in the introductory section, the EMA draft guideline establishes that the parameter of interest to prove quality equivalence is (*µ_T_* − *µ_R_*)/*µ_R_*. As a hypotheses testing problem, to prove equivalence corresponds to rejecting a null hypothesis of non-equivalence, Equation (9):(9)$$H_{0}:\left(\mu_{T}-\mu_{R}\right)/\mu_{R}\leq -0.1\;\mathrm{or}\;\left(\mu_{T}-\mu_{R}\right)/\mu_{R}\geq +0.1$$
for an alternative of equivalence, Equation (10):(10)$$H_{1}:-0.1<\left(\mu_{T}-\mu_{R}\right)/\mu_{R}<+0.1.$$

Provided that (*µ_T_* − *µ_R_*)/*µ_R_* = *µ_T_*/*µ_R_* − 1, these hypotheses may be restated as in Equation (11):(11)$$H_{0}:\mu_{T}/\mu_{R}\leq 0.9\;\mathrm{or}\;\mu_{T}/\mu_{R}\geq 1.1\;\mathrm{vs}.\;H_{1}:0.9<\mu_{T}/\mu_{R}<1.1$$
and the problem is reduced to obtain a 90% confidence interval for the ratio *μ_T_*/*μ_R_*.

Note that these equivalence limits are symmetric around 1. Being limits for a ratio of means it would be more natural to consider asymmetric limits 0.9 to 1/0.9 (=1.1111…), in a similar way to the standard bioequivalence limits 0.8 to 1/0.8 = 1.25. But, for quality equivalence, the draft guideline states the problem in terms of the difference of means relative to the *R* mean and this naturally conducts to the limits stated at (11), which we will consider from now on.

If, according to the guideline, normality is assumed for the corresponding quality measure, the following exact confidence interval with confidence level 1 − 2*α*, for the ratio of means may be derived from the Fieller’s theorem [6], Equation (12):(12)$$\frac{1}{1-g}\left[\frac{{\bar{Y}}_{T}}{{\bar{Y}}_{R}}\pm \frac{t_{r,\alpha }}{{\bar{Y}}_{R}}\sqrt{MS_{B}\left(\frac{1-g}{nb_{T}}+\frac{{\left({\bar{Y}}_{T}/{\bar{Y}}_{R}\right)}^{2}}{nb_{R}}\right)}\right]$$
where $g=\;\;t_{r,\alpha }^{2}MS_{B}/\left(nb_{R}{\bar{Y}}_{R}^{2}\right)$ and *t_r_*_,_*_α_* stands for the critical positive value of the central Student’s ***t*** distribution with *r* = *b_T_* + *b_R_* − 2 degrees of freedom which leaves a probability 2*α* outside the limits $\pm t_{r,\alpha }.$
Appendix A outlines the proof of these results, jointly with a detailed illustrative example of the computations.

##### Direct Confidence Interval for the Difference of Means

The statement defining the quality equivalence criterion, i.e., (10), may be restated as $-0.1\mu_{R}<\mu_{T}-\mu_{R}<+0.1\mu_{R}$. Thus, a simpler but conceptually less correct alternative approach would be to obtain a confidence interval for the difference of means and (by directly substituting the unknown *μ_R_* by its estimate, ${\bar{Y}}_{R}$) to conclude equivalence if it is fully included in the (random) limits $\pm 0.1{\bar{Y}}_{R}$. In other words, quality equivalence may be concluded if the condition in Equation (13) is met:(13)$$-0.1\;{\bar{Y}}_{R}<\left({\bar{Y}}_{T}-{\bar{Y}}_{R}\right)\pm t_{r,\alpha }\sqrt{MS_{B}\left(\frac{1}{nb_{T}}+\frac{1}{nb_{R}}\right)}<0.1\;{\bar{Y}}_{R}.$$

Except for the confidence intervals under consideration, this equivalence criterion is the same as those discussed in the previous subsection. Note that by dividing all terms by ${\bar{Y}}_{R}$, the expression (13) is reduced to requiring that the following confidence interval for the relative difference of means, (*µ_T_* − *µ_R_*)/*µ_R_*, Equation (14),
(14)$$\frac{\left({\bar{Y}}_{T}-{\bar{Y}}_{R}\right)}{{\bar{Y}}_{R}}\pm \frac{t_{r,\alpha }}{{\bar{Y}}_{R}}\sqrt{MS_{B}\left(\frac{1}{nb_{T}}+\frac{1}{nb_{R}}\right)}=\left(\frac{{\bar{Y}}_{T}}{{\bar{Y}}_{R}}-1\right)\pm \frac{t_{r,\alpha }}{{\bar{Y}}_{R}}\sqrt{MS_{B}\left(\frac{1}{nb_{T}}+\frac{1}{nb_{R}}\right)}$$
must be fully included between the ±0.1 limits. After some elementary algebraic manipulation, the above criterion is reduced itself to the condition that a confidence interval for the ratio *µ_T_*/*µ_R_* must be fully included in the range 0.9 to 1.1.

Though common in applications, hereinafter we will not consider this approach because these confidence intervals are not completely accurate, as they fail to take into account the variability in the *μ_R_* estimation at their denominator.

##### Confidence Interval for Log-Transformed Data

Although the draft guideline states clearly that normality can be assumed for the quality measures, arguments were provided at the EUFEPS open forum [7] concerning this assumption. If assuming normality at log-scale were more plausible, the ratio of means previously considered should now be viewed as a ratio of geometric means at the original log-normal scale, which becomes a difference of means at the log-transformed scale. Then the formulation effect, *ϕ* = *ϕ_T_* − *ϕ_R_*, which is the mean difference between both forms at logarithmic scale (now all parameters and their estimates refer to these log transformed data), can be unbiasedly estimated by $\hat{\phi }={\bar{Y}}_{T}-{\bar{Y}}_{R}.$

The meaning of all previous symbols remains the same, but is considered or computed for the log-transformed variables. The variance of $\hat{\phi }$ is, Equation (15):(15)$$\left(n\sigma_{B}^{2}+\sigma^{2}\right)\left(\frac{1}{nb_{T}}+\frac{1}{nb_{R}}\right)$$
which can be directly estimated as in Equation (16):(16)$$\hat{\mathrm{var}}\left(\hat{\phi }\right)=MS_{B}\left(\frac{1}{nb_{T}}+\frac{1}{nb_{R}}\right)$$
and then the procedure may be stated as follows:Logarithmically transform the quality variable under consideration. Now *Y* in model (4) stands for the log-transformed values.With these log-values, compute a 1 − 2*α* confidence interval (typically a 90% interval, with *α* = 0.05) for the form effect, *ϕ*, at logarithmic scale, Equation (17):
(17)$$\hat{\phi }\mp t_{r,\alpha }\sqrt{MS_{B}\left(\frac{1}{nb_{T}}+\frac{1}{nb_{R}}\right)}.$$Back-transform this confidence interval to obtain a 90% (or more in general 1 − 2*α*) for the geometric means ratio between *T* and *R* on the original scale, Equation (18):
(18)$$IC_{GMR}=\exp\left\{\hat{\phi }\mp t_{r,\alpha }\sqrt{MS_{B}\left(\frac{1}{nb_{T}}+\frac{1}{nb_{R}}\right)}\right\}.$$
Equivalence is declared if *IC_GMR_* is fully included within the equivalence limits 0.9 to 1.1 (or any other limits which were considered adequate), Equation (19):
(19)$$0.90<\exp\left\{\hat{\phi }\mp t_{r,\alpha }\sqrt{MS_{B}\left(\frac{1}{nb_{T}}+\frac{1}{nb_{R}}\right)}\right\}<1.1.$$



Otherwise it is not possible to declare equivalence.

#### 2.4.3. Coping with Quality Parameter Multiplicity

For multivariate concepts like rheology, separately testing and proving equivalence for all intervening (correlated) variables, one by one, is unfeasible. One possibility is to summarize them in just one or a few variables. The most obvious possibility is a linear combination, giving appropriate weights to the original variables. These coefficients or weights may be fixed in advance by experts, which could be an arduous task, prone to subjectivity—and even harder if the initial election of variables is not always the same.

As a more feasible approach, provided that all variables under consideration are quantitative, we suggest summarizing the information of all these (continuous) variables by means of the well-known technique of principal component analysis, PCA (for example, see Johnson et al. [8]). In other words, we recommend obtaining the weighting coefficients from the sample information itself. As a result, the starting variables will be linearly combined in a single variable (the first principal component, *P*_1_), or a few variables (the first two or three principal components) which capture most of the information about them. As a final step, the univariate approach for bioequivalence, described in the preceding subsections, will be applied to this or these uncorrelated summary variables. The interest of the analyses based on the principal components may be reinforced if they are interpretable as representing concepts in topic products quality assessment, but such interpretations correspond to the experts in this area, not to statisticians.

Let *P*_1_, *P*_2_, …, *P_k_* be the principal components obtained from the diagonalization of the sample correlations matrix between the original variables. As is well known, these principal components are linear combinations of the form *P_i_* = *a_i_*_1_*X*_1_ + *a_i_*_2_*X*_2_ + … + *a_ik_X_k_*. The *X_j_* stand for the original variables but rescaled to have unit variance so that they are fully comparable, independently of arbitrary choices like the units in which the original variables are expressed. As an example, if *Y* stands for a rheological concept like “initial viscosity”, *X* may correspond to $\left(Y-\bar{Y}\right)/S,$
$\bar{Y}$ may correspond to its sample mean and *S* to its sample standard deviation. The weights *a_ij_* are coefficients verifying $\sum_{j=1}^{k}a_{ij}^{2}=1$. The principal components are non-correlated variables capturing the variability of the original variables in decreasing order of importance, Equation (20):(20)$$\begin{array}{l} \mathrm{var}\left(P_{1}\right)+\mathrm{var}\left(P_{2}\right)+\ldots +\mathrm{var}\left(P_{k}\right)=k=\overset{=1}{\overset{︷}{\mathrm{var}\left(X_{1}\right)}}+\overset{=1}{\overset{︷}{\mathrm{var}\left(X_{2}\right)}}+\ldots +\overset{=1}{\overset{︷}{\mathrm{var}\left(X_{k}\right)}}\\ \mathrm{var}\left(P_{1}\right)\geq \mathrm{var}\left(P_{2}\right)\geq \ldots \geq \mathrm{var}\left(P_{k}\right). \end{array}$$

How many components to use as an adequate summary of the original variables is a matter of discussion. Among others, one may use the common but arbitrary criterion of using the first *h* ≤ *k* components in order to ensure a minimum of an 80% of cumulative variance, or simply just the first component as the one capturing most (but enough?) variability of the original, standardized variables—a choice which would be in the same vein as using a given combination decided in advance by experts. There are many methods whose goal is to decide (in a more objective way) the most convenient number of principal components to consider. Peres-Neto et al. [9] performed a complete comparative study of them. Besides the convenience of a significant result on the sphericity test of Jackson-Bartlett [10] (in other words, to confirm the existence of a non-null correlation structure on the input variables) as a previous step before performing a PCA and determining the number of principal components to consider, they concluded that there is not a single optimal solution for such a determination, because their performance greatly depends on the underlying correlation structure of the variables under consideration. Thus, solutions will be inevitably ad-hoc, for each individual case.

For equivalence testing purposes, the principal components as outlined before are not directly adequate as a summary of the original variables. Intuitively, the coefficients *a_ij_* provide the appropriate weights to construct a good summary, but the variances of the principal components are on a very different scale (they add up to *k*, the number of variables) compared to the original variables, *Y_i_*. As a consequence, each one of the principal components used in the equivalence analyses, *P_i_*, *i* = 1,…, *h*, should be rescaled to be on the same scale as the original variables. Provided that the variance of a linear combination of *Y*_1_, …, *Y_k_*, with weights *a_i_*_1_, …, *a_ik_*, has variance $\sum_{j=1}^{k}a_{ij}^{2}\mathrm{var}\left(Y_{j}\right)$, to be used for equivalence testing purposes, each *P_i_* should be rescaled multiplying it by the factor at Equation (21):(21)$$\left(P_{i}/S_{P_{i}}\right)\sqrt{\sum_{j=1}^{k}a_{ij}^{2}S_{j}^{2}},$$
where $S_{P_{i}}$ stands for the sample standard deviation of the *N* observed values of *P_i_* and $S_{j}^{2}$ for the sample variance computed from the *N* observed values of each initial variable *Y_j_*.

It should also be remembered that in the process of finding the principal components, the original variables were also centered, and PCA was performed on variables having all zero means. So adding the combined mean: $\sum_{j=1}^{k}a_{1j}{\bar{Y}}_{j}$ would translate the first principal component to a location comparable with of the original variables.

This will provide the scores *p*_1_ to be analyzed for equivalence, in the same way as any original variable. All equivalence analysis methods discussed above are applicable, but care must be taken with the confidence interval (18): As a PCA artifact, some principal components may have all their scores negative, which will result in a numeric error. To avoid this problem, simply take their absolute values, the ratios and variances will remain the same. This is the only step where a log-transformation is provided for. In particular, remember that the input original variables are standardized but never log-transformed.

## 3. Results

Presented first are the results of the equivalence analyses for the Kern Pharma generic product vs. the reference Leo Pharma Daivodet reference product. These analyses were performed separately for each one of the quantitative quality characteristics described in the Methods section, and also summarizing the multivariate rheology characteristics by means of a reduced number of principal components. This section ends with the same equivalence analyses but comparing two packaging formats of the same reference product. These additional results are subsequently used to discuss whether the first proposal of equivalence limits is too stringent or not.

### 3.1. Equivalence Analysis for Quantitative Quality Characteristics: Test vs. Reference

Considering the design and data characteristics described previously, all analyses were based on two test batches, *b_T_* = 2, and 10 reference batches, *b_R_* = 10, nested into the respective levels of the “form” factor. Of these 10 reference batches, 5 corresponded to the 30 g commercial packaging format and 5 to the 60 g format. This distinction will be considered at the end of the Results section, which deals with the equivalence analysis of two subsets of the same reference product.

Admittedly, the EMA draft proposal states a minimum of three batches per form, but the development of the generic product was discontinued. This low number of test batches would penalize the power of the equivalence tests of the reference form vs. the test form. For each batch, a balanced number of *n* = 12 replicated observations were obtained. Then, all the equivalence tests described in this section were based on the following fixed quantities (introduced in Section 3.2): *r* = *b_T_* + *b_R_* − 2 = 2 + 10 − 2 = 10 degrees of freedom and *t_r_*_,_*_α_* = *t*_10,0.05_ = 1.8125 critical Student’s *t* value. These values are the same for each particular quality variable and for any measurement scale: original or logarithmic.

#### 3.1.1. Equivalence Analysis of Spreadability (AUC): Test vs. Reference

At the original scale of the “area under the curve”, *AUC*, the test and reference sample mean values are ${\bar{Y}}_{T}=330781$ and ${\bar{Y}}_{R}=342224,$ respectively, and then the sample ratio point estimate is 0.9666, near to one which intuitively suggests equivalence. This is corroborated by the Fieller’s confidence interval, [0.9182, 1.01572], which is fully included inside the 0.9 to 1.1 equivalence limits. Thus, equivalence with respect to *AUC* can be concluded. The details of the computation of this confidence interval can be found in Appendix A.

#### 3.1.2. Equivalence Analysis of the Rheological Parameters: Test vs. Reference

Presented first are the equivalence analyses for each one of the seven rheological parameters cited previously. Considering that the calculation process is the same as before, for the sake of the brevity the results are summarized in Table 1, which also shows the equivalence final outcome for less restrictive limits: ±20% (i.e., 0.8 to 1.2 for the ratio of means) and ±25% (i.e., 0.75 to 1.25).

The results in Table 1 illustrate that the declaration of equivalence for all the parameters under consideration is very unlikely, even if less restrictive equivalence limits were used. A more feasible approach would be to summarize the information regarding these correlated rheological parameters in a few *uncorrelated* components and to perform the equivalence analysis on them.

The sphericity test of Bartlett-Jackson [10] on the above seven rheological variables results in a very high chi-square statistic value, 1306.16, with 24 degrees of freedom. Thus, it makes sense to proceed with a PCA.

According to the (arbitrary) 80% criterion, the first three principal components seem to provide an adequate summary of the seven rheological parameters (the percentages of cumulative variance are 44.73, 66.33, and 80.66, respectively, in a PCA analysis for the standardized variables). On the other hand, the more rigorous criterion of Horn [11] based on comparing the experimental principal component structure with spherical randomly generated structures, retains only the first two components—and this will be also our choice. 

As stated previously, before performing their equivalence analysis, the scores for each principal component must be rescaled to make them comparable in variability with the initial variables. Once rescaled, the equivalence analysis may proceed in a similar way to the one described before but for the principal component scores.

Table 2 displays the estimated mean and variances of the seven original rheological variables under consideration.

The estimated standard deviation of the first principal component scores is 1.7756. We also have $\sum_{j=1}^{k}a_{1j}{\bar{Y}}_{j}=-294263.9$ and $\sum_{j=1}^{k}a_{1j}^{2}S_{j}^{2}=835512327$. After rescaling the first principal component scores according to (20) and submitting them to equivalence analysis, we have that the test and reference means are ${\bar{Y}}_{T}=-292237.7$ and ${\bar{Y}}_{R}=-294669.2,$ respectively, and the estimated ratio is 0.9917. The 90% confidence interval (12) for the means ratio at original scale is [0.8562, 1.1336]. It would not be possible to declare equivalence for the first principal component, if the regulatory equivalence limits 0.9 to 1.1 were used, although the conclusion would be favorable to equivalence for wider limits such as 0.8 to 1.2.

A similar conclusion (no declaration of equivalence) should be reached for the second principal component, provided that the corresponding confidence interval is [1.1410, 1.2947] (with a negative result also for the wider limits being considered).

### 3.2. Equivalence in Quality Analysis: Two Packaging Formats of the Reference Product

Finally, Table 3 presents the results when two packaging formats, 30 g and 60 g, are analyzed for equivalence. In principle, it might seem reasonable to assume that proving equivalence would be easier in this case than when comparing test and reference products. However, the two sets of analysis results are not fully comparable. The main drawback of the test vs. reference analyses was the low number of test batches and its great unbalancing (*b_T_* = 2, *b_R_* = 10), whereas in this analysis we had batch balancing although the total number of batches was lower (*b_30g_* = 5, *b_60g_* = 5).

According to the sphericity test [10] (chi-square statistic of 1105.69 with 21 degrees of freedom) it is worth summarizing the rheological variables by means of a PCA. Method [11] suggests retaining the first two principal components.

## 4. Discussion

In this study, we focused on the consequences of the draft guideline on quality and equivalence of topical products [1], from a statistical point of view. A case-study is presented, in which a new semi-solid product developed by Kern Pharma was analyzed for quality equivalence vs. the Leo Pharma Daivodet reference product.

This study reflects on and provides solutions to the usual statistical methods required to establish equivalence when there is a batch factor, potentially unbalanced, and when the equivalence must be established between multiple variables.

A new statistical method to establish equivalence, based on the Fieller’s theorem, is proposed. This method is more exact than the usual approach of dividing a confidence interval for the difference of means by the estimate of the reference mean, which does not take into account the variability in the reference mean estimate. The Fieller’s confidence interval (12) is mathematically exact (if the assumption on normality of the corresponding quality variable, at its original scale, is adequate) but its mathematical expression is somewhat cumbersome, although the authors would provide R code to compute it, on demand. If, instead, the assumption of normality is more appropriate for log-transformed data, the simpler and more standard confidence interval (19) must be recommended.

The batch factor is also considered. Batch variability makes declaring equivalence in quality slightly more difficult than if the batch factor is not taken into account, but our conclusion is that its omission is a methodological error.

For the equivalence limits stated in the draft guideline, to demonstrate equivalence between *T* and *R* seems as difficult as demonstrating equivalence in terms of physical properties between two sets of batches of *R*, namely, batches corresponding to one packaging form vs. batches corresponding to another packaging form. These results suggest that the equivalence limits of the European draft guideline (±10% of the reference mean) are too restrictive, and make declaring equivalence very difficult, even in cases where it is expected. In this comparison of packaging forms, it is worth pointing out that the reduced number of batches being considered (a total of 10, with 5 batches for each packaging form) may negatively impact on the possibility of declaring equivalence, but this tendency is partly offset by balancing (12 batches, 2 vs. 10 when *T* and *R* are compared). This feeling of excessive restrictiveness is further accentuated by the foreseeable high number of (correlated) quality parameters that will be tested: there will be a high likelihood of being unable to declare equivalence for one or more of them, unless some summarizing strategy is adopted, such as PCA, as suggested here.

Using PCA is not exempt of limitations. As has been stated previously, its main unresolved problem concerning the applicability of this approach is to choose an adequate number of principal components to be tested for equivalence. Choosing too few components can lead to ignoring important information on topical products quality, and choosing too many can introduce noise, analyzing irrelevant variables. In this study we finally adopted partially ad-hoc solutions like testing to discard sphericity (in other words, to provide some evidence on the adequacy of performing PCAs) and subsequently applying the method of Horn [11] to decide the number of relevant principal components.

Finally, it is worth emphasizing that equivalence studies of topical products will presumably encounter high variability issues, especially if batch variability is included in the analyses (as it should be). High variability will also be expected under our suggestion of summarizing by means of PCA. This subject is not considered here, but in our opinion the final guideline should take into account the possibility of some scaling of the equivalence limits, or any other approach that can cope with this high variability.

## Figures and Tables

**Table 1 pharmaceutics-12-00318-t001:** Equivalence analysis of the generic form (Test) vs. brand product (Reference). Each rheological parameter is treated separately.

				±10% Limits (Regulatory), i.e., 0.9 to 1.1 for the Ratio of Means		±20% Limits, i.e., 0.8 to 1.2 for the Ratio of Means		±25% Limits, i.e., 0.75 to 1.25 for the Ratio of Means	
Variable	Estimated T/R Ratio of Means	Confidence Interval for the Ratio of Means		Estimated T/R Ratio of Means Inside Equivalence Limits?	Confidence Interval Inside Limits (Declare Equivalence)?	Estimated T/R Ratio of Means Inside Equivalence Limits?	Confidence Interval Inside Limits (Declare Equivalence)?	Estimated T/R Ratio of Means Inside Equivalence Limits?	Confidence Interval Inside Limits (Declare Equivalence)?
Relative thixotropy, *Sr*	1.4811	**1.2901**	**1.6884**	no	no	no	no	no	no
Yield stress, *σ*_0_	0.9575	**0.8015**	**1.1218**	yes	no	yes	yes	yes	yes
Zero Shear Viscosity, *η*_0_	0.9549	**0.8298**	**1.0853**	yes	no	yes	yes	yes	yes
Viscosity at 100s^−1^, *η*_100_	0.8407	**0.7500**	**0.9340**	no	no	yes	no	yes	no
Viscoelastic modulus, *G′*	1.0161	**0.8253**	**1.2200**	yes	no	yes	no	yes	yes
Viscous modulus, *G″*	1.0117	**0.8531**	**1.1889**	yes	no	yes	yes	yes	yes
Loss tangent, tan(*δ*)	0.9958	**0.9582**	**1.0340**	yes	yes	yes	yes	yes	yes

**Table 2 pharmaceutics-12-00318-t002:** Estimated variance of the original variables.

Variable	Relative Thixotropy, *Sr*	Yield Stress, *σ*_0_	Zero Shear Viscosity, *η*_0_	Viscosity at 100 s^−1^, *η*_100_	Viscoelastic Modulus, *G′*	Viscous Modulus, *G″*	Loss Tangent, tan(*δ*)
Mean	36.511	515.011	625333.2	9.3719	52336.60	36501.26	0.6992
Variance	68.249	3642.392	5186393796	0.8421	59769561	23124700	0.000386

**Table 3 pharmaceutics-12-00318-t003:** Equivalence analysis between two packaging formats of the brand product (Reference).

				±10% Limits (Regulatory), i.e., 0.9 to 1.1 for the Ratio of Means		±20% Limits, i.e., 0.8 to 1.2 for the Ratio of Means		±25% Limits, i.e., 0.75 to 1.25 for the Ratio of Means	
Variable	Estimated T/R Ratio of Means	Confidence Interval for the Ratio of Means		Estimated T/R Ratio of Means Inside Equivalence Limits?	Confidence Interval Inside Limits (Declare Equivalence)?	Estimated T/R Ratio of Means Inside Equivalence Limits?	Confidence Interval Inside Limits (Declare Equivalence)?	Estimated T/R Ratio of Means Inside Equivalence Limits?	Confidence Interval Inside Limits (Declare Equivalence)?
Extensibility, *AUC*	0.9670	**0.9273**	**1.0079**	yes	yes	yes	yes	yes	yes
Relative thixotropy, Sr	0.9198	**0.7899**	**1.0644**	yes	no	yes	no	yes	yes
Yield stress, σ_0_	1.1274	**1.0127**	**1.2539**	no	no	yes	no	yes	no
Zero Shear Viscosity, η_0_	1.1552	**1.0729**	**1.2434**	no	no	yes	no	yes	yes
Viscosity at 100s^−^^1^, η_100_	0.9318	**0.8801**	**0.9855**	yes	no	yes	yes	yes	yes
Viscoelastic modulus, G′	1.2564	**1.1642**	**1.3560**	no	no	no	no	no	no
Viscous modulus, G″	1.2035	**1.1075**	**1.3076**	no	no	no	no	yes	no
Loss tangent, tan(δ)	0.9573	**0.9379**	**0.9769**	yes	yes	yes	yes	yes	yes
Principal component 1	1.1982	**1.1474**	**1.2513**	no	no	yes	no	yes	no
Principal component 2	0.9202	**0.4580**	**1.6451**	yes	no	yes	no	yes	no

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
