# Peer review of "Statistical Methods for Quality Equivalence of Topical Products. 0.5 mg/g Betamethasone Ointment as a Case-Study"

_pharmaceutics, 2020, doi:10.3390/pharmaceutics12040318_

Round 1
Reviewer 1 Report
In general:
The topic of the paper is highly relevant! As the authors express, further discussion and clarifications may be needed in the EMA-guideline with regard to the mentioned issues, in order to enable quality bioequivalence testing. Otherwise the EMA-guideline might be little helpful for the industry, and the marketing of affordable safe generic drugs may be hampered in Europe.
Minor issues/questions, and ideas regarding the problem to solve:
Line 292, "Confidence interval for log-transformed data":
When the interval is back-transferred from log to linear, the range may be 0.9 to 1.11 (i.e. not symmetric), in analogy to 0.8 to 1.25 for BE. Currently the sentence says "0.9 to 1.1", but the sentence in 292 is not clear whether log or linear is meant, in this section on the interval from log-transformed data.
Line 155: I'm not sure if the word "effort" is ideal in this context of physics.
Line 374: "relegated to Appendix A" may be hard to understand for non-natives. "provided in Appendix A" would be easier.
Line 437: If "R and T are compared" is turned around to "T and R..", it would better reflect the underlying mathematics (T vs R, or T/R).
Lines 437 to 442: A thought/idea: If each test parameter out of a growing number of parameters needs to fullfil the acceptance criteria(+-10%), the chance of a successful result (BE) tends to ZERO. This is what the current sentence says, and it is correct. When I now think of the concept of Bonferroni (of multiple testing of parameters, parameters in addition to the primary-aim i.e. strictly hypothesis-based parameters): Wouldn't the implementation of the guideline need a "reversed-Bonferroni" criterium, i.e. widening of the acceptance interval with the number of test parameters? It would be similar to the "Scaled Average Bioequivalence Approach - SABE" currently used for IVPT-data of higher variation, but here the approach would (also) need to scale the interval size with the number of test parameters. If the idea/analogy is helpful, it could be included. If not, please just explain to me.
Author Response
In general:
The topic of the paper is highly relevant! As the authors express, further discussion and clarifications may be needed in the EMA-guideline with regard to the mentioned issues, in order to enable quality bioequivalence testing. Otherwise the EMA-guideline might be little helpful for the industry, and the marketing of affordable safe generic drugs may be hampered in Europe.
Minor issues/questions, and ideas regarding the problem to solve:
Line 292, "Confidence interval for log-transformed data":
When the interval is back-transferred from log to linear, the range may be 0.9 to 1.11 (i.e. not symmetric), in analogy to 0.8 to 1.25 for BE. Currently the sentence says "0.9 to 1.1", but the sentence in 292 is not clear whether log or linear is meant, in this section on the interval from log-transformed data.
Note: All modifications derived from the comments reviewer 1 are highlighted in yellow colour on the revised paper.
Answer: This is a good point. In our opinion, following the guideline at face value (expressions 9 to 11) the ±0.1 limits are specified for the difference of means relative to the R mean, at original scale. Then, the end conclusion is that the limits for the ratio of means at original scale must be 0.9 to 1.1, irrespective of which confidence interval (direct confidence interval for the ratio at original scale or back-transformed interval for log-transformed data) is used to decide equivalence. This is in contrast with the standard criterion for bioequivalence, which is stated directly for the ratio of geometric means and conducts to natural, non-symmetric limits 0.8 to 1/0.8 = 1.25 for this ratio, also at original scale, or ±0.223 for the difference of means at log scale.
Admittedly, limits 0.9 to 1/0.9 = 1.11 (or more precisely 1 / 0.9 = 1.11111…) would be more dependable if the problem was stated directly on ratio of means grounds (as in “classical” bioequivalence) at original scale, and ±0.10 (or more precisely ±0.1054) at log scale for the difference. In other words, we also prefer the limits 0.9 to 1.11 but we think that the limits 0.9 to 1.1 must be used if the draft guideline problem statement is applied verbatim.
We have added a short explanation of these facts just after expression (11).
Line 155: I'm not sure if the word "effort" is ideal in this context of physics.
Answer: It has been changed by “strength”.
Line 374: "relegated to Appendix A" may be hard to understand for non-natives. "provided in Appendix A" would be easier.
Answer: Changed to “can be found in Appendix” following the advice of an English native proofreader.
Line 437: If "R and T are compared" is turned around to "T and R..", it would better reflect the underlying mathematics (T vs R, or T/R).
Answer: Changed.
Lines 437 to 442: A thought/idea: If each test parameter out of a growing number of parameters needs to fulfil the acceptance criteria(+-10%), the chance of a successful result (BE) tends to ZERO. This is what the current sentence says, and it is correct. When I now think of the concept of Bonferroni (of multiple testing of parameters, parameters in addition to the primary-aim i.e. strictly hypothesis-based parameters): Wouldn't the implementation of the guideline need a "reversed-Bonferroni" criterium, i.e. widening of the acceptance interval with the number of test parameters? It would be similar to the "Scaled Average Bioequivalence Approach - SABE" currently used for IVPT-data of higher variation, but here the approach would (also) need to scale the interval size with the number of test parameters. If the idea/analogy is helpful, it could be included. If not, please just explain to me.
Answer: It is an interesting idea but we think that it needs more elaboration before applied. For the moment, we feel unable to elaborate it in the short term. Note that the method of Bonferroni, and similar methods to cope with testing multiplicity, are mainly devices to protect decision criteria (in the present setting, to declare or not equivalence) from committing type I errors (to protect from user risk), i.e., here to declare equivalence when in fact it does not hold. We still don’t see a way to reverse them to protect against type II errors, i.e., not declaring equivalence when in fact it holds (and still preserving from type I error at the same time).
The reviewer also mentions SABE where the limits are widened, scaling them in function of variability. It would be worth considering this possibility, perhaps in conjunction with our suggestion of applying PCA. Principal components concentrate the variability of the original variables. This may lead to high variability problems and thus to low power.
We feel that all these are interesting suggestions, but possibly the subject of a brand new paper.

Reviewer 2 Report
This is an interesting work introducing an attempt to rationalize equivalence criteria for topical drugs.
Although personally I don’t believe this approach could be adopted by Regulatory, I congratulate Authors for starting discussion on how to simplify this approach and therefore I find this paper worthy publication. I’ve got few points:
Why not to use classical but Fieller’s interval? A classical confidence interval is basic for bioequivalence studies. Moreover, Fieller’s CI is based on certain assumptions on means for Test and Reference formulations, thus are not completely universal and in order to implement them require testing of these conditions. Please justify your choice as in my opinion it is not an optimal one even though is exact from the mathematical point of view.
What about logarithmic transformation when using PCA? Do we need to calculate log-parameters before or after PCA?
The number of principle components covering pre-defined amount of variance is data-dependent, thus one cannot assume that it will substantially reduce computational burden for final decision on the products equivalence. What would be the Authors’ recommendation for the “sound” number of PCs in this method? You just comment on certain “expert” knowledge. Please elaborate this angle
Author Response
This is an interesting work introducing an attempt to rationalize equivalence criteria for topical drugs.
Although personally I don’t believe this approach could be adopted by Regulatory, I congratulate Authors for starting discussion on how to simplify this approach and therefore I find this paper worthy publication. I’ve got few points:
Why not to use classical but Fieller’s interval? A classical confidence interval is basic for bioequivalence studies. Moreover, Fieller’s CI is based on certain assumptions on means for Test and Reference formulations, thus are not completely universal and in order to implement them require testing of these conditions. Please justify your choice as in my opinion it is not an optimal one even though is exact from the mathematical point of view.
Note: All modifications derived from the comments of reviewer 2 are highlighted in green colour in the revised version of the paper.
Answer: We agree with the reviewer in that the original paper by Fieller is very general in its scope (and mathematically rather complex). But we are using the simplified version of Fieller’s results which are commonly used in bioequivalence, e.g., in standard textbooks like Chow, S-C. and Liu, J-P.(2009). “Design and Analysis of Biovailability and Bioequivalence Studies”. Chapman & Hall, pp. 88-92, which only requires standard assumptions of normality. In the design that we are considering, these requirements are further simplified by the independence between the sample means of T and R. Then, the required assumptions match with those stated in the draft guideline, (normality at original scale) which does not explicitly call for their verification. Admittedly, the computation of the Fieller’s confidence interval is rather involved, and possibly a classical confidence interval (similar to those used in standard bioequivalence studies) would be preferable if it were applicable. In our opinion, the choice would depend on assuming normality for the original variables (as seems the choice in the draft guideline) or for their log transformation. In the first case, according to the guideline a direct confidence interval for the ratio is required and thus the Fieller’s method introduced in section 2.4.2.1 would be preferable, because it is exactly correct. Alternatively, a standard confidence interval for the mean of T but divided by the sample mean of R (section 2.4.2.2) would be also applicable, but it has the drawback of not taking into consideration the variability in this estimate of the R mean. In the second case (assuming normality at log-scale, not stated in the draft guideline), a classical confidence interval for the difference of means but back-transformed to the original scale (section 2.4.2.3) would be the right choice.
We have added a new paragraph at the Discussion section (from line 451), trying to clarify this point.
What about logarithmic transformation when using PCA? Do we need to calculate log-parameters before or after PCA?
We appreciate this comment; this point was unclear and it is important.
The PCA is performed over the original variables, without any log-transform. The PCA returns one or more new variables, the principal components, which may be analysed for equivalence in the ways discussed in the paper, i.e., either directly by means of the Fieller’s confidence interval or applying the confidence interval in formula (18) after log-transforming the principal component scores. (In the latter case, for one or more components all scores can be negative, which is an arbitrary artefact of the method, but this does not affect the relevant computations like the ratio of means at original scale; then, simply take their absolute value before applying logarithms.) In the examples, the PCAs were performed without any previous log-transformation of the input variables and the equivalence analysis on the resulting principal component scores were performed using the Fieller’s confidence interval, without any log-transformation on them.
We have added some text trying to clarify how this process is performed, from line 358.
The number of principle components covering pre-defined amount of variance is data-dependent, thus one cannot assume that it will substantially reduce computational burden for final decision on the products equivalence. What would be the Authors’ recommendation for the “sound” number of PCs in this method? You just comment on certain “expert” knowledge. Please elaborate this angle
We greatly appreciate this comment which, in our opinion, has contributed to a relevant improvement on the paper.
The reviewer points the greatest problem in PCA applications, which also concerns our proposal to cope with multiplicity in quality parameters. We have elaborated more this subject, adding explanation and analyses at different places, from lines 334, 404, 409, 438 and in the Discussion section, from line 473. (And additional bibliography, line 540.) This issue still remains as an unsolved problem as, unfortunately, we are bound to recognize finally.

Round 2
Reviewer 2 Report
No more comments. Like I wrote before, I don't have to agree with all claims of the Authors but I respect that they took this subject and I believe they should follow it further. Congrats!